# Protocol for a scoping review to identify research reporting on eating disorders in minority ethnic populations in the UK, Canada, Australia and New Zealand

Helena Tuomainen ![ORCID],[1] Rose McGowan,[1] Aliyah Williams-Ridgway,[1] Katie Guy,[2] Sheryllin McNeil[2]

¹University of Warwick, Coventry, UK
²Birmingham Women's and Children's Hospitals NHS Foundation Trust, Birmingham, UK

**Correspondence to**
Dr Helena Tuomainen;
helena.tuomainen@warwick.ac.uk

## ABSTRACT

**Introduction** Eating disorders (EDs) are common, severe and often life-threatening psychiatric conditions. Notwithstanding stereotypes, EDs affect individuals from all racial and ethnic backgrounds. However, despite similar and in some cases increased prevalence of disordered eating and EDs among minority ethnic groups, there appear to be disparities when it comes to ED diagnosis and treatment. To date, most of the existing literature exploring disordered eating and EDs among minority ethnic groups has been conducted in the USA. The present scoping review aims to examine the extent, range and nature of research activity into disordered eating and EDs in the UK, Canada, Australia and New Zealand providing a comprehensive overview of the existing literature. A special focus will be placed on studies exploring prevalence, access to care and treatment experiences.

**Methods and analysis** The scoping review framework first outlined by Arksey and O'Malley and improved on by Levac *et al* was used to guide the development of this scoping review protocol. A detailed systematic search of relevant databases (Medline, EMBASE, PsycINFO, CINAHL and Web of Science) will be conducted. Relevant literature will also be identified from the reference lists of included studies. Two reviewers will independently screen titles and abstracts and full-texts against specified inclusion and exclusion criteria. A third reviewer will resolve disagreements if necessary. Relevant data will be extracted using a data charting form. Quantitative and qualitative summaries of extracted data will be provided.

**Ethics and dissemination** No ethical approval is required for this study. Findings will be of benefit to researchers, clinicians and policy-makers by highlighting areas for future research and identifying ways to making ED treatment more accessible for individuals from all backgrounds. Findings will be disseminated via conferences, presentations and peer-reviewed journals.

## INTRODUCTION
### Background

Eating disorders (EDs) are complex psychiatric conditions characterised by disordered eating and/or weight-control behaviours,[1] which typically have their onset during adolescence.[2] A recent systematic review reported a

---

**STRENGTHS AND LIMITATIONS OF THIS STUDY**

⇒ This protocol will ensure transparency in methodology to reduce the likelihood of reviewing bias.
⇒ Studies conducted in non-Western, low-income or middle-income countries will not be included in this review.
⇒ There will be no quality assessment of the included studies.
⇒ Only studies published in English will be included.

---

pooled global lifetime and 12-month prevalence of EDs of 0.91% and 0.43%, respectively.[3] EDs have profound physical and psychological impacts on the affected individual and those around them. EDs have high comorbidity with other psychiatric conditions and are associated with suicidal thoughts and attempts.[4 5] The medical complications that often accompany EDs involve almost all organ systems and can be life-threatening,[6] leading to significantly increased mortality rates.[7] Given the negative consequences of EDs, it is imperative that we understand their prevalence and treatment within and across ethnic groups.

Despite the prevailing stereotype that EDs only affect 'skinny, White, affluent girls' (the SWAG stereotype),[8] it is widely reported that EDs affect individuals from all racial and ethnic backgrounds.[9] However, findings regarding the prevalence of disordered eating and EDs across ethnic groups are mixed. Some studies report no difference in disordered eating and ED prevalence across different ethnic groups.[10–12] Others, however, have reported higher prevalence of disordered eating among minority ethnic men and women relative to White individuals.[13] In contrast, some studies have found higher prevalence of EDs in White adults when compared with other racial and ethnic

groups.[8 14 15] Reported prevalence seems to vary based on the racial or ethnic group and type of disordered eating or ED being studied.[9] Differences in study design and diagnostic criteria used may also contribute to differences in reported prevalence.[11]

Despite similar and in some cases increased rates of disordered eating and EDs among minority ethnic groups there appear to be disparities when it comes to ED diagnosis and treatment. In their systematic review, Sinha and Warfa[16] found minority ethnic individuals in the UK and the USA were less likely to seek and receive ED treatment as well as less frequently diagnosed and referred to specialist ED services. Since publication of this review, further studies have replicated these findings.[8 17 18] Such disparities in diagnosis and treatment have been attributed to a lack of perceived need for treatment among these populations,[8] differences in symptom presentation[18] and culturally insensitive diagnostic criteria.[19]

To date, most of the existing literature exploring disordered eating and EDs among minority ethnic groups has been conducted in the USA. This is apparent in recent narrative and systematic reviews which have explored the identification, prevalence, presentation, risk factors and treatment of disordered eating and EDs among minority racial and ethnic populations.[9 16 20–23] The overwhelming majority of studies included in these reviews have been conducted in the USA.

Researchers have called for more research exploring ethnic and racial diversity in EDs to be conducted outside of the USA.[24] The generalisability of findings from studies conducted in the USA to countries with different healthcare systems and sociocultural and environmental contexts has been questioned.[11 25] Franko and Rodgers[26] also argue research outside of the USA is necessary to fully understand the extent of similarities and differences within and across ethnic groups.

### Rationale

The present scoping review aims to identify and describe studies reporting on disordered eating and EDs in minority ethnic groups in the UK, Canada, Australia and New Zealand; to our knowledge, this will be the first review to do so. These countries have large established minority ethnic populations due to high levels of immigration and/or the forced displacement of Indigenous peoples, as well as broadly comparable healthcare systems. By providing a broad overview of the extent, range and nature of existing research conducted in these countries we hope to elucidate any knowledge gaps to inform future research. This is crucial considering the identified need for more research exploring racial and ethnic differences in EDs outside of the USA.[24]

Given the well-documented racial and ethnic health disparities in the diagnosis and treatment of EDs[16] and mental health conditions more broadly,[27] a special focus will be placed on studies which have reported on the prevalence of disordered eating and EDs, access to care and treatment experiences among minority ethnic

groups. Such studies will be particularly relevant in identifying any health disparities in the countries of focus and highlighting possible ways to overcome these. This is of particular importance given prompt diagnosis and intervention can improve ED treatment outcomes and recovery prospects.[28]

### Objectives

In this scoping review, we aim to examine the extent, range and nature of research activity into disordered eating and EDs among minority ethnic populations in the UK, Canada, Australia and New Zealand. The specific objectives are to:
1. Map the literature according to the main themes/topics addressed, target populations and methodologies used.
2. Report on findings regarding ED prevalence, access to care and treatment experiences.
3. Identify gaps and/or limitations in the literature and suggest areas for future research.

## METHODS AND ANALYSIS
### Protocol design

A scoping review was chosen as the most appropriate methodology. Scoping reviews are suited to answering broad research questions and are designed to identify the types of evidence available in a field, examine how this research has been conducted and highlight any knowledge gaps.[29] The present scoping review will follow the five-stage framework first outlined by Arksey and O'Malley[30] and subsequently improved by Levac et al.[31] The stages are as follows: (1) identifying the research questions; (2) identifying the relevant literature; (3) selecting studies; (4) charting the data and (5) collating, summarising and reporting the results. To ensure the quality of the scoping review, the development and reporting of this protocol followed the Preferred Reporting Items for Systematic Reviews and Meta-Analyses Extension for Scoping Reviews (PRISMA-ScR) guidelines.[32] The protocol for this scoping review is also registered on OSF (https://osf.io/w3d6g). Any changes made as part of the iterative nature of the scoping review will be documented here.

### Stage 1: identifying the research questions

The first stage of a scoping review is to develop research questions which are informed by the purpose and objectives of the scoping review.[31] Based on the aforementioned objectives of this scoping review, we have formulated six research questions. These are described in further detail below. Given the iterative nature of scoping reviews, a reflexive approach will be adopted. If necessary, as we become more familiar with the existing literature our research questions will be refined and modified accordingly.

### Question 1: what topics are being addressed by the current literature?

Within the USA, studies have explored the identification, prevalence, presentation, risk factors and treatment of EDs among minority racial and ethnic groups. For each study included in this scoping review, we will report on the primary focus of the study. By doing so, we hope to provide a summary of the themes/topics that have been addressed by the current literature on disordered eating and EDs among minority ethnic populations in the UK, Canada, Australia and New Zealand.

### Question 2: how has this research been conducted?

This research question aims to explore how the included studies have been designed and conducted. For each study, we will report the research design and methodology used to answer research questions.

### Question 3: which populations have been targeted for this research?

By answering this question, we hope to provide an overview of which groups and populations have been studied. We will report on key participant characteristics (e.g., age, gender, socioeconomic status, comorbidities) and examine which ethnic groups and types of disordered eating or EDs have been studied. We also aim to explore how and where participants were recruited (e.g., sampling technique, clinical or non-clinical sample, sample size).

### Question 4: are there differences in reported ED prevalence across ethnicity?

This question aims to assess whether reported prevalence rates of disordered eating and EDs differ across ethnic groups.

### Question 5: are there differences in access to ED treatment across ethnic groups?

To answer this question, we will extract and analyse data from studies which have reported findings regarding any of the following: referrals to ED services, service utilisation, barriers to care or help-seeking (or treatment seeking).

### Question 6: do ED treatment experiences differ across ethnic groups?

The purpose of this question is to assess what is currently known regarding the treatment experiences of minority ethnic patients. We will aim to explore how minority ethnic individuals and/or stakeholders involved in the delivery or receipt of their treatment (e.g., clinicians, family members) describe their experiences of treatment. We will also aim to identify the types of treatment that have been used in minority ethnic populations.

## Stage 2: identifying the relevant literature
### Information sources

The second stage of a scoping review is to identify relevant studies for selection and data extraction. This will involve a systematic search of the following electronic databases: (1)

Medline, (2) PsycINFO, (3) EMBASE, (4) CINAHL and (5) Web of Science. To ensure the search is as comprehensive as possible, backwards citation chaining of included studies will be used to identify any additional studies of relevance. For the sake of completeness, a web search using Google Scholar will also be completed. Both published and grey literature (e.g., theses) will be considered for inclusion.

### Search strategy

The search strategy was developed by members of the research team (HT, SM and RM) in collaboration with a speciality academic support librarian. Search terms include keywords and terms related to (1) EDs and (2) ethnicity. See online supplemental appendix 1 for the search strategy for each of the databases. No date limits will be applied to the search results, but the database search query will be set to produce articles in English only due to resource constraints.

The search results for each database will be documented and references will be imported into the citation management system EndNote, where duplicates will be removed before studies are screened for inclusion.

## Stage 3: selecting studies

The process of selecting studies will consist of two stages: (1) title and abstract screening and (2) full-text screening. Each stage will involve two reviewers independently screening the identified studies against the inclusion and exclusion criteria. All screening will be conducted using the online screening tool Rayyan.

In the first round of screening, two reviewers (RM and HT) will review paper titles and abstracts. Papers will be categorised as either 'include', 'exclude' or 'maybe'. Decisions as to whether papers are eligible for full-text review will be made as follows: (1) if both reviewers agree to 'include', the study will move on to the second screening; (2) if both reviewers agree to 'exclude', the study will not be read in full and will be excluded from the review and (3) if the reviewers are not unanimous in their decisions, the study will move onto the second screening to be read in full before a final decision is made. Where a paper abstract is unavailable, papers will be assigned to the 'maybe' category and automatically move on to the second stage of screening to be read in full. Eligibility criteria will be further refined and finalised following the first round of screening if necessary.

In the second round of screening, two reviewers (AW-R and HT) will read the remaining articles in full to determine whether they meet the inclusion criteria. If the decision to include or exclude is not unanimous, both reviewers will assess the study together and discuss disagreements. A third reviewer (SM) may be consulted for arbitration of any conflicting decisions. A flow chart outlining decisions made regarding study inclusion will be produced using the PRIMSA template.[33]

### Eligibility criteria

Eligibility criteria were developed using the Population, Concept, Context framework[34] (see table 1). Inclusion

**Table 1** Core elements of the scoping review based on Population Concept, Context framework

| Population | Minority ethnic groups | A group of people within a country that: 'is numerically smaller than the rest of the population; is not in a dominant position; has a culture, language, religion or race that is distinct from that of the majority; and its members have a will to preserve those characteristics'. (Foa as cited in United Nations p.97 [37] [38]). |
|---|---|---|
| Concept | Eating disorders | Clinical psychiatric disorders, including anorexia nervosa, bulimia nervosa and binge eating disorder, characterised by disturbed eating behaviours, body image concerns and unhealthy weight control behaviours.[1] |
| | Disordered eating | Subclinical levels of unhealthy eating behaviours including restrictive eating, binge eating and unhealthy weight control behaviours.[39] |
| Context | | Studies conducted in the UK, Canada, Australia or New Zealand. |

and exclusion criteria are further detailed below (see table 2).

### Stage 4: charting the data/data extraction

Once studies have been identified for final inclusion, data from these studies will be extracted and charted. A comprehensive data charting form will be developed in Excel and used to extract relevant information from studies. Two reviewers (AW-R and HT) will pilot the data charting form on a sample of randomly selected papers (10% of included papers). Any discrepancies or uncertainties highlighted by this exercise will be discussed and the charting form will be adjusted accordingly. One reviewer (AW-R) will extract the data from the remaining papers. The research team will meet regularly throughout the data extraction process to discuss progress and further changes to the data extraction form will be made as needed in an iterative process. For example, additional categories of interest may emerge. Any changes made and the rationale for doing so will be reported in final scoping review findings.

Data to be extracted will include some standard information about the paper. For example, study title, author(s), year of publication and country of publication. Additional information specific to our research questions will also be extracted. This will include the study aims/objectives, research focus/topic, study design and methodology, sample characteristics and demographics (e.g., age, gender, ED diagnosis, ethnicity, sample size), setting in which the study was conducted (e.g., clinical or non-clinical setting), summary of relevant findings and recommendations of the author(s) if applicable. In line with guidance on conducting scoping reviews, there will be no critical appraisal of the included studies.[32]

### Stage 5: collating, summarising and reporting the results

Rather than meta-synthesis or critical appraisal of studies, the purpose of a scoping review is to provide an overview of the existing literature, summarising what is already known

**Table 2** Inclusion and exclusion criteria

| | Inclusion | Exclusion |
|---|---|---|
| Population | ► Subsample or full sample of minority ethnic participants or data related to minority ethnic individuals.<br>► Where multiple ethnic groups are included in a sample the findings are reported according to ethnicity. For example, ethnicity is the main predictor variable in analyses or the paper reports on ethnic differences in study outcomes (e.g., prevalence rates, scores on psychometric measures, referral rates etc.).<br>► All ages | ► Where multiple ethnic groups are included in the sample, but authors did not report any outcomes or analysis according to ethnicity (e.g., ethnicity only reported in sample demographics, analyses reported for sample as a whole rather than individual ethnic groups, where analyses are controlled or adjusted for ethnicity but no further findings or data relating to ethnicity are reported). |
| Concept | ► Study focuses on clinical EDs (i.e., meeting DSM or ICD criteria as assessed using clinician interview or a validated psychometric measure) or disordered eating (i.e., assessed with a validated psychometric measure). | ► Focuses solely on risk factors, associated concepts or related conditions (e.g., food addiction, obesity, body image).<br>► Focuses solely on ARFID, pica or rumination disorder. |
| Context | ► Study conducted in UK, Canada, Australia or New Zealand. | |
| Study design | ► Any study design (e.g., case reports, case studies, cohort studies, randomised controlled trials). | |
| Publication type | ► Peer-reviewed journal articles<br>► Theses | ► Review articles, editorials, commentaries, letters, newsletters, opinion or reflection pieces, book chapters, book reviews, poster presentations or conference abstracts. |
| Language | ► Published in English | |

ARFID, Avoidant Restrictive Food Intake Disorder; DSM, Diagnostic and Statistical Manual of Mental Disorders; ED, eating disorder; ICD, International Classification of Diseases .

and identifying any knowledge gaps.[29] The extracted data will be summarised in a way that maps the extent, range and nature of research into disordered eating and EDs among minority ethnic populations in the UK, Canada, Australia and New Zealand, while placing particular focus on ethnic differences in prevalence, access to care and treatment experiences.

We will provide a descriptive numerical summary of the characteristics of included studies. Studies will be classified and grouped according to their characteristics. Extracted data will be presented in tables, charts and visual maps. To accompany tabular and visual summaries, we will also provide a qualitative narrative summary of the data and how it relates to our research questions. If data and resources permit, qualitative findings relating to treatment experiences of minority ethnic individuals will be synthesised following Thomas and Harden's[35] thematic synthesis approach. Thematic synthesis adapts the key principles of thematic analysis[36] for use in reviews. The aim of the analysis is to identify and interpret themes/patterns across papers. This is done across three stages (1) line-by-line coding of the results section; (2) organising codes into descriptive themes and (3) generation of analytical themes.[35] If conducted, further details on this analysis procedure will be provided in the final review. Finally, informed by any significant gaps and/or limitations of the literature, we will recommend avenues for future research. The scoping review findings will also be discussed in relation to clinical practice and policy. Reporting of the results will follow PRISMA ScR guidelines.[32]

## ETHICS AND DISSEMINATION

As scoping reviews involve the methodological examination and integration of existing literature and resources, ethical approval is not required. Once all stages of the scoping review are complete, findings will be disseminated via conferences, presentations and the submission of the final review for peer-reviewed publication.

### Patient and public involvement

Patients or the public have not been involved in the design, conduct, reporting or dissemination plans of this scoping review.

**Acknowledgements** The authors thank Samantha Johnson, Academic Support Librarian, for assisting in the search strategy.

**Contributors** HT and SM were involved in the initial conception of the review. HT is responsible for the design of the review. RM aided with the design of the review and drafted the first manuscript. AW-R revised the manuscript making critical edits. KG contributed meaningfully to the drafting and editing of the manuscript. All authors read and approved the final manuscript.

**Funding** This research is funded by the National Institute for Health and Care Research (NIHR) Applied Research Collaboration (ARC) West Midlands (grant number NIHR200165).

**Disclaimer** The views expressed are those of the authors and not necessarily those of the NIHR or the Department of Health and Social Care.

**Competing interests** None declared.

**Patient and public involvement** Patients and/or the public were not involved in the design, or conduct, or reporting, or dissemination plans of this research.

**Patient consent for publication** Not applicable.

**Provenance and peer review** Not commissioned; externally peer reviewed.

**ORCID iD**
Helena Tuomainen http://orcid.org/0000-0003-1636-8187

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
