## [Reviewer comments · BMJ Open]

ARTICLE DETAILS

TITLE (PROVISIONAL)	A protocol for a scoping review to identify research reporting on eating disorders in minority ethnic populations in the UK, Canada, Australia, and New Zealand
AUTHORS	Tuomainen, Helena; McGowan, Rose; Williams-Ridgway, Aliyah; Guy, Katie; McNeil, Sheryllin

VERSION 1 – REVIEW

REVIEWER	Burnette, Blair Michigan State University, Psychology
REVIEW RETURNED	27-Jul-2023

GENERAL COMMENTS	I appreciate the opportunity to review this protocol for a scoping review of the relation between eating disorders and ethnicity. For far too long, the ED literature has suffered from a focus on white, cisgender, young, thin women. This focus has harmed outcomes for those who don't fit the SWAG stereotype. So, I applaud the authors for attempting to address the limitations imposed on the literature by this focus. Yet, I think this protocol needs revision, especially regarding the clarity and concreteness of the research aims. I elaborate on my feedback below. Overarching concern: -Throughout the manuscript, the language is often vague and would benefit from being more concrete and direct. I will offer specific examples below. Multiple times I had to re-read sentences and passages to read between the lines or figure out exactly what the authors planned to do. I think scientific articles are already at great risk of not being read and disappearing into the void. As researchers, I think one of the best things we can do is make it easy for our readers. Stay away from abstraction and vague text and instead be direct and concise. Minor concerns: -This manuscript would benefit from thorough proofreading as there are numerous typographic errors (e.g., p. 4, "there a number...") -Be consistent in whether you capitalize race/ethnicity (e.g., "White" vs. "white") throughout the manuscript Introduction -In the second sentence, the authors note prevalence data from a recent systematic review-can the authors clarify if the review assessed global ED prevalence? -Minor point, but some reorganization of the second paragraph may be helpful. The line, "Similarly, some researchers have found..."
--

	seems to go after the sentence ending with, "...in comparison to White peers" as they both concern prevalence. It seems tangential to have a sentence about referrals to ED services between two sentences about prevalence. -I recommend against referring to people as their racial/ethnic group (e.g., "Blacks"). Race/ethnicity is an adjective. For more, please see: https://www.hamilton.edu/academics/centers/writing/writing-resources/writing-about-race-ethnicity-social-class-and-disability https://usca.edu/icb/training-resources/guide-to-inclusive-language/inclusive-language-guide/file -The second full paragraph on p. 4 ("There have been a number...") is when things begin to get overly vague for me. For instance, "practical barriers" (what are these?), "negative treatment attitudes" (meaning what?), "lack of support and encouragement" (from whom?). -Again, the Rationale paragraph on p. 4 is vague. For instance, "the disparity in current literature surrounding EDs and ethnic groups..." What does this mean? Disparity in what? Do you mean the equivocal findings regarding the prevalence of EDs? The lack of data we have on how EDs present across groups? Why we see differences in treatment-seeking and diagnosis? Further, I think the term, "ethnic variation" is tripping me up, because again, it's vague. I think this review is trying to capture a lot of information - I understand it's a scoping review, but the scope is huge. You're attempting to look at ED prevalence across and within ethnicity, differences in treatment-seeking, and potentially differences in treatment outcomes. Therefore, it's likely challenging to succinctly state the paper's purpose. Yet, I think a more precise term than "ethnic variation" needs to be used. Can the authors re-write the rationale paragraph to be more precise and concrete regarding the study aims? -Similarly in the objectives, I don't think "Conduct a systematic search of literature concerning eating disorders and ethnicity" is sufficiently clear. That could mean anything! Methods -The authors again state that they'll report on the "ethnic variations" in the prevalence, access to services, and experience of treatment in eating disorders. Could the authors rephrase this, perhaps saying that they will assess whether there are differences across ethnicity in reported ED prevalence, access to services (how will this be measured? Is this actually what you're measuring or will you be looking at reported engagement with treatment? Or referrals? Or seeking treatment?), and treatment outcomes? Are you actually going to be looking at how people across racial/ethnic groups describe their treatment experiences? Or are you looking at outcomes data? -Earlier in the Methods, I would emphasize this is a global review that is not restricted by country except that only articles in English will be included. -Question 2) "What types of research have been conducted?" - What do you mean by types? Methodologies?
--	---

	-I am skeptical that the authors will be able to assess treatment effectiveness by ethnic group. At least in the US, RCTs of eating disorder treatment have not provided sufficiently granular data to assess treatment efficacy within or between groups (see Burnette et al., 2022). I am curious how the authors will overcome the limitations of data reported within treatment trials to assess this -This is a small point, but I don't think identifying literature gaps needs to be a primary research question, because inherently the other research questions will likely elucidate many gaps -The authors include a PRISMA checklist for scoping reviews in the supplemental material, but I would state in the body of the manuscript how you plan to integrate this tool -Inclusion/exclusion criteria: again, these are vague and ill-defined. For instance, what does "Reports on EDs and ethnicity" entail? How will you assess that it is doing that? What does it mean for a paper to present findings by ethnic group? What if the authors do an omnibus significance test but adjust for race/ethnicity? Would this count? -"Ethnic minority of that country" - there is insufficient detail here. What does it mean that an inclusion criteria of a study is "ethnic minority of that country"? -Given the reported inclusion/exclusion criteria ("Does not include findings from ethnic minorities"), would a paper on ED prevalence in Africa be included if it does not report on the prevalence of EDs among white participants? I would think including studies from countries where the ethnic majority group has been historically excluded from ED research would be important. -Are the authors also planning to preregister this review? I appreciate submitting this protocol for review, but pre-registering would offer authors the chance to provide updates on the 'iterative' process they describe in the protocol.
--	---

VERSION 1 – AUTHOR RESPONSE

Comment	Author Response
Overarching Concern	
1. Throughout the manuscript, the language is often vague and would benefit from being more concrete and direct. I will offer specific examples below. Multiple times I had to re-read sentences and passages to read between the lines or figure out exactly what the authors planned to do. I think scientific articles are already at great risk of not being read and	The authors would like to thank the reviewer for their time and effort spent reviewing our scoping review protocol. Their detailed and constructive comments have been valuable. Whilst editing the manuscript an effort has been made to take into consideration the comments regarding the lack of clarity in the writing. We have aimed to use direct and unambiguous language, offer clearer explanations of what we intend to

disappearing into the void. As researchers, I think one of the best things we can do is make it easy for our readers. Stay away from abstraction and vague text and instead be direct and concise.	do and improve the specificity of the scoping review objectives and research questions.
Minor Comments	
2. This manuscript would benefit from thorough proofreading as there are numerous typographic errors (e.g., p. 4, "there a number...")	The manuscript has been proofread by the authors and any typographic errors have been corrected.
3. Be consistent in whether you capitalize race/ethnicity (e.g., "White" vs. "white") throughout the manuscript	In accordance with guidelines on reporting on race and ethnicity, all racial and ethnic groups have been capitalised (e.g., "Asian", "Black" and "White" etc.) and capitalization is consistent throughout the manuscript.
4. In the second sentence, the authors note prevalence data from a recent systematic review-can the authors clarify if the review assessed global ED prevalence?	This sentence has been edited to make it clear to the reader that the prevalence estimates reported are referring to global eating disorder prevalence.
Introduction	
5. Minor point, but some reorganization of the second paragraph may be helpful. The line, "Similarly, some researchers have found..." seems to go after the sentence ending with, "...in comparison to White peers" as they both concern prevalence. It seems tangential to have a sentence about referrals to ED services between two sentences about prevalence.	The structure of introduction section has been reorganized. As suggested all studies relating to prevalence are described before moving on to discuss evidence relating to referrals and access to treatment.
6. I recommend against referring to people as their racial/ethnic group (e.g., "Blacks"). Race/ethnicity is an adjective.	Terminology used when writing about race and ethnicity has been amended in accordance with published guidelines. Where mentioned race and ethnicity are used in adjectival rather than noun form.
7. The second full paragraph on p. 4 ("There have been a number...") is when things begin to get overly vague for me. For instance, "practical barriers" (what are these?), "negative treatment attitudes" (meaning what?), "lack of support and encouragement" (from whom?).	The introduction has been rewritten and revised. As one of several changes to make the introduction more concise the specific paragraph referred to in this comment has been deleted.
8. Again, the Rationale paragraph on p. 4 is vague. For instance, "the disparity in current literature surrounding EDs and ethnic groups..." What does this mean? Disparity in what? Do you mean the equivocal findings regarding	The rationale paragraph has been rewritten to provide clarification as to what the scoping review aims to achieve. The primary aim is to identify and describe studies reporting on disordered eating and eating disorders in minority ethnic groups in the UK,

the prevalence of EDs? The lack of data we have on how EDs present across groups? Why we see differences in treatment-seeking and diagnosis? Further, I think the term, "ethnic variation" is tripping me up, because again, it's vague. I think this review is trying to capture a lot of information - I understand it's a scoping review, but the scope is huge. You're attempting to look at ED prevalence across and within ethnicity, differences in treatment-seeking, and potentially differences in treatment outcomes. Therefore, it's likely challenging to succinctly state the paper's purpose. Yet, I think a more precise term than "ethnic variation" needs to be used. Can the authors re-write the rationale paragraph to be more precise and concrete regarding the study aims?	Canada, Australia and New Zealand. Thus, giving a broad overview of the extent, nature and range of existing research. A particular focus will be placed on prevalence, access to care and treatment experiences, with secondary objectives being to answer questions relating to differences in these areas across ethnicity.
9. Similarly in the objectives, I don't think "Conduct a systematic search of literature concerning eating disorders and ethnicity" is sufficiently clear. That could mean anything!	The objective to "conduct a systematic search of literature concerning eating disorders and ethnicity" has been removed. We recognise this objective is ambiguous and could be interpreted in several ways. Further, systematic literature searching is intrinsic to the scoping review methodology and so need not be listed as a specific objective. The remaining objectives of the scoping review have been revised to improve clarity and explicitly convey what we as authors intend to achieve.
Methods	
10. The authors again state that they'll report on the "ethnic variations" in the prevalence, access to services, and experience of treatment in eating disorders. Could the authors rephrase this, perhaps saying that they will assess whether there are differences across ethnicity in reported ED prevalence, access to services (how will this be measured? Is this actually what you're measuring or will you be looking at reported engagement with treatment? Or referrals? Or seeking treatment?), and treatment outcomes? Are you actually going to be looking at how people across racial/ethnic groups describe	The research questions for the scoping review have been revised and rewritten. We have tried to be as clear as possible with regards to what each of our questions is asking and have provided information regarding how we intended to answer each question.

their treatment experiences? Or are you looking at outcomes data?	
11. Earlier in the Methods, I would emphasize this is a global review that is not restricted by country except that only articles in English will be included.	After further reading of relevant literature, a preliminary review of initial search results and receiving reviewer feedback we have made the decision to narrow the focus of our scoping review. We will do this by restricting inclusion by country and focusing on studies conducted in the UK, Canada, Australia and New Zealand. The volume of literature focusing on eating disorders within minority ethnic groups was larger than initially expected. We therefore recognised that a global review would be a substantial undertaking and not feasible given our current time and resource constraints. The majority of research exploring eating disorders among minority ethnic populations has been conducted in the United States, this is apparent in recent relevant narrative and systematic reviews (for example see Acle et al., 2021; Goode et al., 2020; Rodgers et al., 2018; Sinha & Warfa, 2013) which include mostly US studies. Indeed, there have been calls for more research exploring racial and ethnic differences in eating disorders to be conducted in other countries. We hope our scoping review will be able to provide a broad overview of the nature and type of research that has been conducted in the countries of focus and identify priority research areas. Our countries of interest were selected on the basis that they have established minority ethnic populations and broadly comparable healthcare systems.
12. Question 2) "What types of research have been conducted?" - What do you mean by types? Methodologies?	This research question has been reframed as "How has this research been conducted?". This research question aims to explore what methodologies were used by included studies to answer their research questions.
13. I am sceptical that the authors will be able to assess treatment effectiveness by ethnic group. At least in the US, RCTs of eating disorder treatment have not provided sufficiently granular data to assess treatment efficacy within or between groups (see Burnette et al., 2022). I am curious how the authors	Rather than assessing treatment effectiveness by ethnicity, in this scoping review we will focus on the treatment experiences of minority ethnic patients. The research question "Do eating disorder treatment experiences differ across ethnic groups?" aims to explore how people across different racial and ethnic groups (and/or relevant stakeholders involved in treatment) describe their experiences of treatment. We will

will overcome the limitations of data reported within treatment trials to assess this.	also aim to identify what types of treatment have been used with minority ethnic patients (i.e. descriptions of these treatment).
14. This is a small point, but I don't think identifying literature gaps needs to be a primary research question, because inherently the other research questions will likely elucidate many gaps.	At the suggestion of the reviewer, identifying research gaps being is no longer listed as a primary research question.
15. The authors include a PRISMA checklist for scoping reviews in the supplemental material, but I would state in the body of the manuscript how you plan to integrate this tool.	Additional information has been provided in the methods and analysis section on how the PRISMA ScR checklist has been used in the present scoping review (i.e. development of protocol and reporting of the scoping review).
16. Inclusion/exclusion criteria: again, these are vague and ill-defined. For instance, what does "Reports on EDs and ethnicity" entail? How will you assess that it is doing that? What does it mean for a paper to present findings by ethnic group? What if the authors do an omnibus significance test but adjust for race/ethnicity? Would this count? "Ethnic minority of that country" - there is insufficient detail here. What does it mean that an inclusion criteria of a study is "ethnic minority of that country"?	Inclusion and exclusion have been redefined to provide further clarification as to which studies will be eligible for inclusion in the review.
17. Given the reported inclusion/exclusion criteria ("Does not include findings from ethnic minorities"), would a paper on ED prevalence in Africa be included if it does not report on the prevalence of EDs among white participants? I would think including studies from countries where the ethnic majority group has been historically excluded from ED research would be important.	Please see above response referring to narrowing the focus of the scoping review by country. Countries where the ethnic majority have historically been excluded from research will no longer be eligible for inclusion in this scoping review. We recognise that eating disorders are not limited to US or other Western countries and the decision to exclude studies conducted in non-Western countries has been highlighted as a limitation of this scoping review. However, this decision was made on the basis of resource and time constraints. Future scoping reviews could focus specifically on non-Western countries, providing a useful overview of the current evidence available within these regions.
18. Are the authors also planning to preregister this review? I appreciate submitting this protocol for review, but	In light of reviewer comments, we have also registered the protocol for this scoping review on OSF. Doing so will allow for the

pre-registering would offer authors the chance to provide updates on the 'iterative' process they describe in the protocol.	documentation and justification of any changes to or deviations from the protocol as we progress into the study selection and data extraction process.
Editor Comments	
19. Please include the planned search dates in the abstract.	Please could you clarify if 'planned search dates' refers to the date the search was executed, or the publication dates searched in databases. Database searches were completed 30 March 2023. No date limits were applied to search results.
20. Please include, as a supplementary file, the precise, full search strategy (or strategies) for all databases, registers and websites, including any filters and limits used.	A supplementary file outlining the full search strategy (including any filters and limits) used for all databases searched.

VERSION 2 – REVIEW

REVIEWER	Burnette, Blair Michigan State University, Psychology
REVIEW RETURNED	02-Jan-2024

GENERAL COMMENTS	The revised manuscript is much improved. I commend the authors for their responsiveness. Overall, I think the manuscript is strengthened and this review will be a great addition to the literature. The only lingering suggestion that I have is that the authors add more detail about the potential thematic analysis, space permitting. Thematic analysis is not a single approach, but a collection of approaches. Right now the methodology for the thematic analysis is poorly elaborated. However, as these analyses may not be conducted depending on the findings of the review, I do not find inclusion of more detailed information absolutely crucial for publication.
---

VERSION 2 – AUTHOR RESPONSE

Reviewer Comments	
1. The revised manuscript is much improved. I commend the authors for their responsiveness. Overall, I think the manuscript is strengthened and this review will be a great addition to the literature. The only lingering suggestion that I have is that the authors add more detail about the potential thematic analysis, space permitting. Thematic analysis is not a single approach, but a collection of approaches. Right now the	The authors would like to thank the reviewer for their positive feedback regarding the updated manuscript. We have added some additional detail as to how the thematic analysis would be conducted. At present it is unknown whether the thematic analysis will be completed, as stated in the protocol this will be dependent upon the studies identified and time resources. If thematic analysis is conducted,

methodology for the thematic analysis is poorly elaborated. However, as these analyses may not be conducted depending on the findings of the review, I do not find inclusion of more detailed information absolutely crucial for publication.	the methodology for the thematic analysis will be outlined in further detail in the final review.
--	--